# Comparison of the clinical performance and usefulness of five SARS-CoV-2 antibody tests

**Mitsuru Wakita[1], Mayumi Idei[2,3], Kaori Saito[2], Yuki Horiuchi[2], Kotoko Yamatani[2], Suzuka Ishikawa[4], Takamasa Yamamoto[1], Gene Igawa[1], Masanobu Hinata[1], Katsuhiko Kadota[5], Taro Kurosawa[6], Sho Takahashi[6], Takumi Saito[7], Shigeki Misawa[1], Chihiro Akazawa[8], Toshio Naito[9], Takashi Miida[2], Kazuhisa Takahashi[10], Tomohiko Ai[2,11]\*, Yoko Tabe[2,12]**

1 Department of Clinical Laboratory, Juntendo University Hospital, Tokyo, Japan, 2 Department of Clinical Laboratory Medicine, Juntendo University Faculty of Medicine, Tokyo, Japan, 3 Medical Technology Innovation Center, Juntendo University Faculty of Medicine, Tokyo, Japan, 4 Tokyo Medical and Dental University School of Health Care Sciences, Tokyo, Japan, 5 Emergency and Disaster Medicine, Juntendo University Faculty of Medicine, Tokyo, Japan, 6 Department of Gastroenterology, Juntendo University Faculty of Medicine, Tokyo, Japan, 7 Department of Internal Medicine and Rheumatology, Juntendo University School of Medicine, Tokyo, Japan, 8 Intractable Disease Research Center, Juntendo University Graduate School of Medicine, Tokyo, Japan, 9 Department of General Medicine, Juntendo University Faculty of Medicine, Tokyo, Japan, 10 Department of Respiratory Medicine, Juntendo University Faculty of Medicine, Tokyo, Japan, 11 Department of Medicine, Indiana University School of Medicine, Indianapolis, Indiana, United States of America, 12 Department of Next Generation Hematology Laboratory Medicine, Juntendo University Graduate School of Medicine, Tokyo, Japan

\* t-ai@juntendo.ac.jp, ait@iu.edu

**Data Availability Statement:** Data contains sensitive identifying information (confidential clinical information) and cannot be shared publicly due to ethical restrictions imposed by Dr. Atsushi Okuzawa, the Chair of Juntendo University Hospital

## Abstract

We examined the usefulness of five COVID-19 antibody detection tests using 114 serum samples at various time points from 34 Japanese COVID-19 patients. We examined Elecsys Anti-SARS-CoV-2 from Roche, and four immunochromatography tests from Hangzhou Laihe Biotech, Artron Laboratories, Chil, and Nadal. In the first week after onset, Elecsys had 40% positivity in Group S (severe cases) but was negative in Group M (mild-moderate cases). The immunochromatography kits showed 40–60% and 0–8% positivity in Groups S and M, respectively. In the second week, Elecsys showed 75% and 50% positivity, and the immunochromatography tests showed 5–80% and 50–75% positivity in Groups S and M, respectively. After the third week, Elecsys showed 100% positivity in both groups. The immunochromatography kits showed 100% positivity in Group S. In Group M, positivity decreased to 50% for Chil and 75–89% for Artron and Lyher. Elecsys and immunochromatography kits had 91–100% specificity. Elecsys had comparable chronological change of cut-off index values in the two groups from the second week to the sixth week. The current SARS-CoV-2 antibody detection tests do not provide meaningful interpretation of severity and infection status. Its use might be limited to short-term epidemiological studies.

Institutional Review Board. Data access requests can be sent to: Dr. Shigeki Aoki, the Chair of Ethical Committee at Juntendo University: E-mail: saoki@juntendo.ac.jp.

**Funding:** Reagents and assays for Elecsys Anti-SARS-CoV-2 were provided by Roche Diagnostics. LYHER Novel Coronavirus (2019-nCoV) IgM/IgG Antibody Combo Test, Artron COVID-19 IgM/IgG Antibody Test, CHIL COVID-19 IgG/IgM Rapid Test, and NADAL COVID-19 IgG/IgM test were provided by Hangzhou Laihe Biotech, Artron Laboratories, Chil, and nal von minden, respectively. All providers did not play any roles in the design of the study, analysis and interpretation of the data.

**Competing interests:** The authors have declared that no competing interests exist.

# Introduction

The new coronavirus disease (COVID-19), caused by severe acute respiratory syndrome coronavirus 2 (SARS-CoV-2), originated from Wuhan, China in late 2019 and spread worldwide. The World Health Organization (WHO) declared the pandemic on March 11, 2020. To control the pandemic, diagnostic tests such as reverse transcription polymerase chain reaction (RT-PCR) methods were developed [1]. The results of these RT-PCR tests were used for taking political decisions such as imposing lockdown in several countries [2, 3]. However, since RT-PCR tests are feasible only within three weeks since symptom onset, it is inconvenient for epidemiological investigations. To estimate past infection numbers, serological tests were developed (https://www.whitehouse.gov/wp-content/uploads/2020/05/Testing-Guidance.pdf). As of January 2021, more than 33 serological tests are commercially available as they were urgently approved by the United States Food and Drug Administration and European Medicines Agency. Importantly, more than 40 serological assays were not approved (https://open.fda.gov/apis/device/covid19serology/), which suggests that the performance of COVID-19 serological assays were not yet thoroughly investigated. In addition, the significance of serological tests remains unclear as the Center for Disease Control published interim guidelines for their use (https://www.cdc.gov/coronavirus/2019-ncov/lab/resources/antibody-tests-guidelines.html).

SARS-CoV-2, a single-stranded RNA virus belonging to the *Orthocoronavirinae* subfamily, consists of four structural components, namely, spike glycoprotein (S), envelope protein, membrane glycoprotein, and nucleocapsid phosphoprotein (N), and 16 non-structural proteins [4]. Thus, the accuracy and reliability of these tests rely upon the nucleotide fragments used to develop the antibody. In addition, viral types may differ across infections at different times. To date, at least 116 mutations including three common mutations have been identified [5], and the seroprevalence timing might differ by viral type.

This study aimed to investigate the sensitivity, specificity, and time course of seroprevalence in 34 Japanese COVID-19 patients using an electrochemiluminescence immunoassay (ECLIA)-based Elecsys Anti-SARS-CoV-2 (RUO, Roche Diagnostics) test and four different immunochromatographic (IC) point-of-care tests developed by Hangzhou Laihe Biotech, Artron Laboratories, Chil, and Nadal.

# Material and methods

## Clinical backgrounds

This study complied with all relevant national regulations and institutional policies and was conducted in accordance with the tenets of the Declaration of Helsinki. The study was approved by the Institutional Review Board (IRB) at Juntendo University Hospital (IRB # 20–036). The need for informed consent from individual patients was waived because all samples were de-identified in line with the Declaration of Helsinki.

Between March and June 2020, 114 serum samples were collected from 34 COVID-19 patients. **Table 1** shows the clinical characteristics and timing of sample collection. All patients were confirmed to be positive according to PCR-based testing of SARS-CoV-2 using the Light Mix Modular SARS-CoV-2 (COVID-19) N-gene and E-gene assay (Roche Diagnostics, Tokyo, Japan) or the 2019 Novel Coronavirus Detection Kit (Shimadzu, Kyoto, Japan). We classified patients into two groups according to the WHO criteria: Group M that included mild and moderate cases and Group S that included severe and critical cases. For the negative control, 100 serum samples collected from outpatients without infectious diseases between November and December 2018 were used. The samples were stored at -80°C until use. All data

**Table 1. Clinical characteristics.**

| | Group M | | | Group S* |
|---|---|---|---|---|
| | **Outpatients** | **Inpatients** | **Total** | |
| Patients number | 16 | 10 | 26 | 8 |
| Female, n (%) | 5 (31.3) | 4 (40.0) | 9 (34.6) | 1 (12.5) |
| Age, year | 43 ± 18 | 51 ± 18 | 46 ± 18 | 70 ± 8 |
| Sample number | 16 | 45 | 61 | 53 |
| 0–6 days** | 12 | 0 | 12 | 5 |
| 7–13 days | 4 | 4 | 8 | 8 |
| 14–20 days | 0 | 13 | 13 | 10 |
| 21–27 days | 0 | 7 | 7 | 7 |
| 28–34 days | 0 | 8 | 8 | 10 |
| 35–41 days | 0 | 9 | 9 | 7 |
| 42- | 0 | 4 | 4 | 6 |

Data are expressed as mean±SD.

*All severe and critical cases were inpatients.

**Days from onset.

were fully anonymized before access, and de-identified clinical information obtained between March and December 2020 were provided.

## Antibody assays

We used the US Food and Drug Administration-approved Elecsys Anti-SARS-CoV-2 electro-chemiluminescence immunoassay system (Roche Diagnostics, Basel, Switzerland), which is based on the modified double-antigen sandwich immunoassay with recombinant nucleocapsid protein (N) and measures SARS-CoV-2 total antibody (pan immunoglobulin) with a fully automated Cobas e801 analyzer (Roche Diagnostics) (https://www.accessdata.fda.gov/cdrh_docs/presentations/maf/maf3358-a001.pdf). According to the FDA, the Elecsys Anti-SARS--CoV-2 system has 100% sensitivity ($\geq$14 days after a positive polymerase chain reaction [PCR] assay) and 99.8% specificity (https://www.fda.gov/medical-devices/coronavirus-disease-2019-covid-19-emergency-use-authorizations-medical-devices/eua-authorized-serology-test-performance). The results are reported as numeric values in the form of a cutoff index (COI; signal sample/cutoff) with qualitative results reactive (COI $\geq$ 1.0; positive). The analytical and clinical performance of the assay have been evaluated and are described elsewhere [6].

The following rapid immunochromatographic IgM/IgG antibody assays were utilized: LYHER Novel Coronavirus (2019-nCoV) IgM/IgG Antibody Combo Test (Hangzhou Laihe Biotech); Artron COVID-19 IgM/IgG Antibody Test (Artron Laboratories); CHIL COVID-19 IgG/IgM Rapid Test (Chil), and NADAL COVID-19 IgG/IgM test (nal von minden). The immunochromatographic IgM/IgG antibody assays target the receptor binding domain of S protein or the nucleocapsid protein, N protein (**Table 2**). The presence of only the control line indicated a negative result, whereas the presence of both the control line and the IgM or IgG antibody line indicated a positive result for IgM or IgG antibody, respectively. **Table 2** summarizes the features of these kits.

## Statistics

Statistical analyses were performed using Stat Flex for Windows (ver. 6.0; Artech, Osaka, Japan). The total Ig index between Group M and Group S was compared using the Mann-Whitney U test. A two-tailed $p$ value of <0.05 was considered statistically significant.

Table 2. Performance specification of reagent and kits.

| Reagent/Kits | Manufacturer | Isotype | Target Protein[*] | Sample volume (μL) | Run time (min) | Approval status [**] |
|---|---|---|---|---|---|---|
| Electrochemiluminescence immunoassay (ECLIA) | | | | | | |
| Elecsys Anti-SARS-CoV-2 | Roche | Total Ig | N | 200 | 18 | FDA (EUA), CE |
| Immnochromatography | | | | | | |
| Lyher novel Coronavirus(2019-nCoV)IgM/IgG Antibody Combo Test | Hangzhou Laihe Biotech | IgM, IgG | S-RBD | 10 | 15 | CE |
| Artron COVID-19 IgM/IgG Antibody Test | Artron Laboratories | IgM, IgG | S-RBD | 10 | 10 | CE |
| Chil COVID-19 IgG/IgM Rapid Test | Chil | IgM, IgG | S-RBD+N | 5 | 15 | CE |
| Nadal COVID-19 IgG/IgM Test | nal von minden | IgM, IgG | S-RBD+N | 10 | 15 | CE |

[*]S-RBD: Receptor Binding Domain of spike protein, N: Nucleocapsid.

[**]FDA (EUD): Food and Drug Administration (Emergency Use Authorization), CE: Conformite Europeenne.

## Results

Table 3 shows the sensitivity or the rate of positivity of Elecsys and the four immunochromatography kits in a total of 114 serum samples from 34 patients. The results of the immunochromatography kits were considered as positive when IgM or IgG were positive (qualitative tests).

In the first week after onset, Elecsys had a 40% positivity in Group S but was negative in Group M. Additionally, the four immunochromatography kits had 40–60% and 0–8% positivity in the Groups S and M, respectively. In the second week, Elecsys showed 75% and 50% positivity in Groups S and M, respectively. The four immunochromatography kits had 63–88% and 25–75% positivity in Groups S and M, respectively. After the third week, Elecsys showed 100% positivity in both groups, except for the fifth week in Group S (90%). Except for Chil, the immunochromatography kits showed 100% positivity in Group S. In Group M, positivity gradually decreased to 50% for Chil (IgM and IgG) and 75–89% for Artron and Lyher. Elecsys and Nadal showed the most consistent positivity.

Specificity was evaluated using the samples collected before the COVID-19 era. Table 4 shows that the specificity of IgM was as low as 91% for Artron and 96% for Nadal. For IgG, all kits showed a specificity of >98%.

### Chronological change of COI

Next, we examined the COI values at various time points after onset using Elecsys. Fig 1 shows that COI tended to increase over time. However, there was no significant difference between Groups M and S until the sixth week. In the seventh week, the COI was higher in Group S than in Group M.

To examine the chronological changes of COI in eight inpatients, the COI values were plotted against the timing of the tests (Fig 2). Table 5 summarizes the patients' clinical background characteristics. Four patients (#1, 6, 7, and 8) required ventilation support, and unfortunately, all patients could not be rescued. Three patients, except patient #1, showed relatively low COIs. The COI of patient #1 reached 100 when the patient died at 52 days. In patient #6, the COI did not increase at 13 days. Importantly, none of the deceased patients showed high COI values on admission. The patients who survived (#2, 3, 4, and 5) received supplemental oxygen and supporting therapies and were eventually discharged. Three of these (#2, 3, and 4) showed relatively high COIs (around 40).

**Table 3. Sensivity of SARS-CoV-2 antibody assay.**

| | Elecsys (Total Ig) | |
|---|---|---|
| | Group M | Group S |
| 0–6 days | 0 | 40 |
| 7–13 days | 50 | 75 |
| 14–20 days | 100 | 100 |
| 21–27 days | 100 | 100 |
| 28–34 days | 100 | 90 |
| 35–41 days | 100 | 100 |
| 42- | 100 | 100 |

| | Lyher | | | | | | Artron | | | | | |
|---|---|---|---|---|---|---|---|---|---|---|---|---|
| | Group M | | | Group S | | | Group M | | | Group S | | |
| | IgM | IgG | IgM/IgG | IgM | IgG | IgM/IgG | IgM | IgG | IgM/IgG | IgM | IgG | IgM/IgG |
| 0–6 days | 8 | 0 | 8 | 60 | 40 | 60 | 8 | 0 | 8 | 60 | 40 | 60 |
| 7–13 days | 50 | 25 | 50 | 75 | 63 | 75 | 63 | 13 | 63 | 75 | 63 | 75 |
| 14–20 days | 100 | 85 | 100 | 100 | 100 | 100 | 100 | 85 | 100 | 100 | 100 | 100 |
| 21–27 days | 100 | 100 | 100 | 100 | 100 | 100 | 100 | 100 | 100 | 100 | 100 | 100 |
| 28–34 days | 100 | 75 | 100 | 100 | 100 | 100 | 100 | 75 | 100 | 100 | 100 | 100 |
| 35–41 days | 100 | 89 | 100 | 100 | 100 | 100 | 100 | 89 | 100 | 100 | 100 | 100 |
| 42- | 100 | 100 | 100 | 100 | 100 | 100 | 100 | 100 | 100 | 100 | 100 | 100 |

| | Chil | | | | | | Nadal | | | | | |
|---|---|---|---|---|---|---|---|---|---|---|---|---|
| | Group M | | | Group S | | | Group M | | | Group S | | |
| | IgM | IgG | IgM/IgG | IgM | IgG | IgM/IgG | IgM | IgG | Ig M/IgG | IgM | IgG | IgM/IgG |
| 0–6 days | 8 | 8 | 8 | 60 | 60 | 60 | 8 | 8 | 8 | 60 | 60 | 60 |
| 7–13 days | 50 | 50 | 50 | 50 | 88 | 88 | 50 | 50 | 63 | 63 | 75 | 63 |
| 14–20 days | 92 | 92 | 100 | 80 | 100 | 100 | 100 | 100 | 100 | 100 | 100 | 100 |
| 21–27 days | 100 | 100 | 100 | 100 | 100 | 100 | 100 | 100 | 100 | 100 | 100 | 100 |
| 28–34 days | 88 | 88 | 100 | 100 | 100 | 100 | 100 | 100 | 100 | 100 | 100 | 100 |
| 35–41 days | 56 | 56 | 89 | 86 | 100 | 100 | 100 | 100 | 100 | 100 | 100 | 100 |
| 42- | 50 | 50 | 100 | 100 | 83 | 100 | 100 | 100 | 100 | 100 | 100 | 100 |

The data were presented as positive result percentage for tested numbers. IgM/IgG indicates positive for either IgM or IgG.

# Discussion

In this study, we evaluated the performance of five different SARS-CoV-2 antibody detection tests using 114 serum samples from 34 Japanese patients with COVID-19 in a Tokyo

**Table 4. Specificity of test kit.**

| | Isotype | Specificity (%) | False positive (%) |
|---|---|---|---|
| Elecsys Anti-SARS-CoV-2 | Total Ig | 99 | 1 |
| Artron COVID-19 IgM/IgG Antibody Test | IgM | 91 | 9 |
| | IgG | 98 | 2 |
| LYHER novel Coronavirus(2019-nCoV)IgM/IgG Antibody Combo Test | IgM | 99 | 1 |
| | IgG | 99 | 1 |
| CHIL COVID-19 IgG/IgM Rapid Test | IgM | 100 | 0 |
| | IgG | 98 | 2 |
| NADAL COVID-19 IgG/IgM Test | IgM | 96 | 4 |
| | IgG | 99 | 1 |

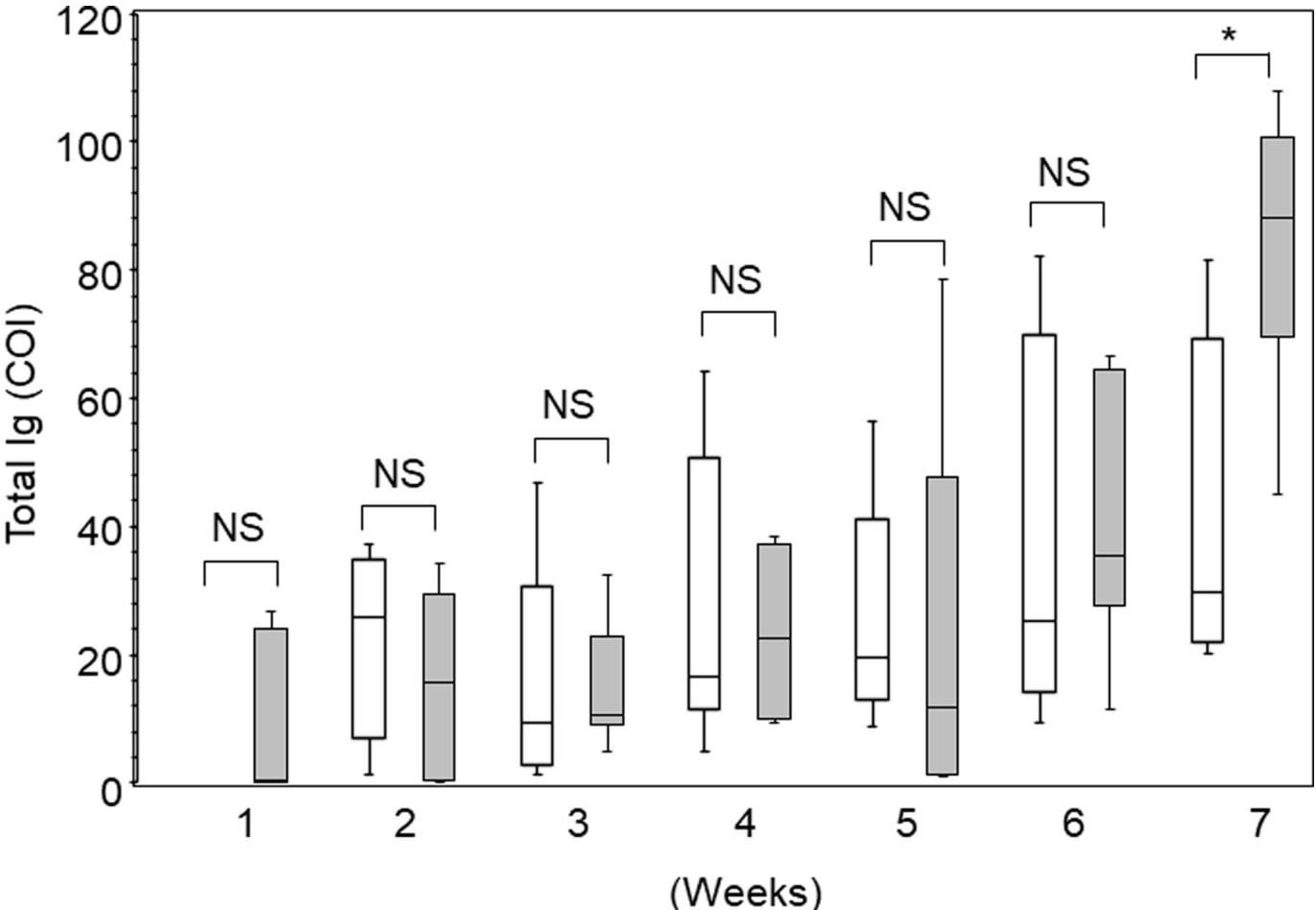

**Fig 1. Seroprevalence of antibodies to SARS-CoV-2.** Antibody Index for SARS-CoV-2 PCR-positive patient samples for the indicated weekly timeframes post-onset of symptoms. The data were presented as mean with interquartile ranges. Open bars indicate Group M, and gray bars indicate Group S. Note that none of Group M showed significant COI values in the first week. $^*p<0.05$; NS, no significant difference.

metropolitan area. Our study demonstrated several important findings. First, the seroprevalence was approximately 40–60% in severe cases and relatively low in mild cases in the first week. The seroprevalence increased to 60–80% in severe cases and 50–60% in mild cases in the second week. After the third week, the seroprevalence reached almost 100% in both groups. In mild cases, the seroprevalence decreased when tested with Artron and Chil kits (**Table 3**). Second, the specificity was not 100% for all tests using the samples collected before the COVID-19 era (**Table 4**). Third, the COI values using Elecsys did not differ significantly over time except for the seventh week (**Fig 1**). However, this might be the effect of one outlier (patient #1 in **Table 5**). In addition, the COI values obtained by Elecsys might not reflect disease severity (**Fig 2**).

It was reported that IgM and IgG could be detected in 20–30% of cases approximately 14 days after onset, and the positive rates reach 80–90% after 15 days [6]. Interestingly, it was reported that IgM and IgG increased almost simultaneously [7]. In currently available antibody detection kits, the antibodies were developed based on the S1 domain of the S protein or the N protein. The N proteins are essential for viral survival and expansion, while the S proteins are essential for binding to the host cell surface receptors [8]. Since the S proteins might be produced before the increase in the N proteins, the performance of antibody detection kits can

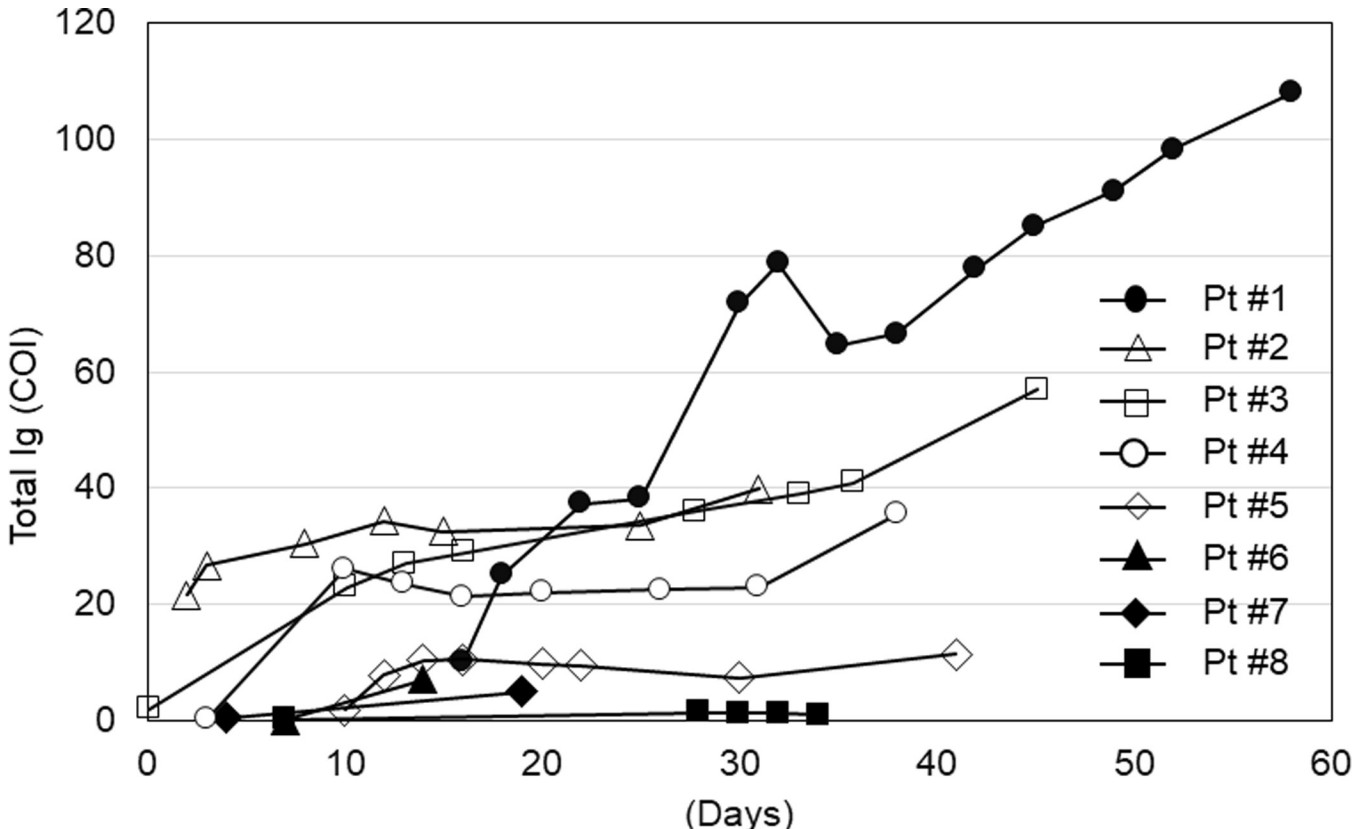

**Fig 2. Longitudinal changes of antibodies against SARS-CoV-2 in severe cases.** The cut-off index in eight severe patients were tested using Elecsys. The COI values were plotted as a function of days after onset. Closed symbols depict deceased cases, and open symbols depict survived cases.

depend upon the target protein of the antibody. This might explain why Chil and Artron kits showed early decline of antibody levels in mild cases. However, how these kits were designed are confidential. Another concern is the false-positive rate of Artron and Nadal kits. Since our negative control samples were collected before 2018, antibodies against SARS-CoV-2 did not

**Table 5. Clinical characteristics of patients with Group S.**

| Patient # | Severity* | Outcome | Age | Sex | Past medical history | Treatments | | |
|---|---|---|---|---|---|---|---|---|
| 1 | Critical | Deceased | 76 | M | Hypertension | Ventilation | Continuous hemodiafiltration |
| | | | | | Diabetes | | Plasmapheresis |
| | | | | | Cancer | | |
| 2 | Severe | Survived | 77 | M | Diabetes | Supplemental oxygen | |
| | | | | | Rheumatoid arthritis | | |
| | | | | | Pneumonia | | |
| 3 | Severe | Survived | 75 | M | Prostatic hypertrophy | Supplemental oxygen | |
| 4 | Severe | Survived | 66 | M | none | Supplemental oxygen | |
| 5 | Severe | Survived | 57 | M | none | Supplemental oxygen | |
| 6 | Critical | Deceased | 78 | F | none | Ventilation | |
| 7 | Critical | Deceased | 64 | F | Hyperlipidemia | Ventilation | Continuous hemodiafiltration |
| | | | | | Cancer | | Plasmapheresis |
| 8 | Critical | Deceased | 67 | M | Hypertension | Ventilation | Continuous hemodiafiltration |
| | | | | | Renal failure | | Plasmapheresis |

exist in these samples. Speculative explanations are antibody purification issues, difference in the target fragments, and crossreaction with other coronaviruses including SARS and Middle East Respiratory Syndrome (MERS).

Currently, SARS-CoV-2 is detected using RT-PCR, and it is believed that SARS-CoV-2 nucleotides can be detected using RT-PCR several days after symptom onset; however, the sensitivity and specificity of this test are unclear [9]. After a certain time period (more than three weeks), the sensitivity of PCR tests declines, and antibody tests may detect antibodies developed against nucleotide fragments of SARS-CoV-2. Currently, except for supporting therapies, there is no available treatment option for COVID-19 despite several cases of experimental drug use in the past several months [10–14]. Moreover, the pattern of seroprevalence remains unclear. Although the sample number was small, the severe cases in our study did not show any meaningful COI changes using Elecsys (**Fig 2**). In addition, our recent study showed that seroprevalence in 4147 healthcare workers in our hospital was 0.34% [doi: 10.21203/rs.3.rs-96870/v1]. Since the prevalence of COVID-19 is largely dependent on the number of PCR tests in a given population [15], it is likely that the prevalence of COVID-19 has been underestimated. Therefore, this suggests that the antibodies detected by current methods might disappear within a short period of time after infection [17].

Although many companies continue releasing new tests, we could test only limited numbers of assays commercially available in Japan when the study was performed. However, studies published in December 2020 have reported varying results in newer tests. Using 36 samples obtained from RT-PCR confirmed COVID-19 patients, Sacristan *et al*. reported that the detection percentage of IgG antibodies were similar in StrongStep SARS-CoV-2 IgG/IgM kit and AllTest COV-19 IgG/IgM kit (83.3%and 80.6%, respectively). In contrast, the IgM detection rates were lower than the IgG detection rates, and different between the two tests (11.1% and 30.6%, respectively) [16]. The timing of the antibody tests was approximately 11 days after RT-PCR tests, which is similar to our results between the second and the third week in Group M. Nilsson *et al*. compared several assays using 98 samples collected at different time points [17]. The assays included: EUROIMMUN anti-SARS-CoV-2 IgG and IgA ELISAs (EUROIMMUN Medizinische Labordiagnostika AG, Lübeck, Germany); WANTAI SARS-CoV-2 IgM ELISA (Beijing Wantai Biological Pharmacy Enterprise, Beijing, China); Acro IgM/IgG Lateral Flow Test (LFT)(2019-nCoV IgG/IgM Rapid Test Cassette, Acro Biotech, Rancho Cucamonga, CA, USA); Livzon IgM/IgG LFT (Diagnostic Kit for IgM/IgG Antibody to Corona Virus, Zhuhai Livzon Diagnostics, Zhuhai, China); and CTK IgM/IgG LFT (OnSiteTM COVID-19 IgG/IgM Rapid Test, CTK Biotech, Poway, CA). According to their results, WANTAI ELISA and Acro LFT were more sensitive than others in detecting IgM antibodies in the first week after onset. However, the sample size was small, consisting of only three patients. WANTAI ELISA and CTK LFT showed higher positivity for IgM between 8 and 28 days, then declined after 28 days. For the IgG antibody detections, all tests showed low sensitivity in the first week. Acro LFT showed a positivity of 91–100% between 8 and 28 days, which was better than the other assays. The other tests showed 57–94% positivity between 8–28 days, then declined after 28 days. They also compared the positivity among the outpatients, hospitalized and ICU admitted patients. All tests tended to show higher positivity in the inpatients compared to the outpatients, which is consistent with our data (**Table 3**). However, the positivity varied depending upon the assay. In addition, a meta-analysis of 57 studies published in June 2020 reported the low sensitivity and high heterogeneity of the serological tests [18]. All these results indicate unreliability and difficulty in developing serological tests against SARS-CoV-2, a single strand RNA virus even with slower mutation rates than other RNA viruses [19, 20]. Furthermore, there are many confounding factors such as difference in methodology, antibody development, and uncertainty of pathogens.

This study has several limitations: (1) this is a single-center study with a relatively small number of patients; (2) since the target nucleotides to develop antibodies are not disclosed, data interpretation was incomplete; (3) since the follow-up time was limited to 42 days, we do not know the long-term detection rate; (4) finally, we do not know whether these antibodies act as neutral antibodies.

In conclusion, our data showed that the serological tests including one ECLIA test and four immunochromatography tests had poor sensitivity during the early phase of infection and therefore were unsuitable for diagnosis or screening. In addition, these tests cannot provide meaningful interpretation of infection status. Thus, the current use of these tests might be limited to short-term epidemiological studies unless newer and more reliable technologies are developed in the future.

## Acknowledgments

The authors are grateful to the participants in this study. The authors also would like to thank Natsumi Itakura, Masayoshi Chonan, Koji Tsuchiya and Takaaki Kawakami for their technical supports, and Dr. Corina Rosales for critical reading manuscript. Finally, we thank to all medical staff who conducted their duties in the treatment of this pandemic.

## Author Contributions

**Conceptualization:** Yoko Tabe.

**Data curation:** Mitsuru Wakita, Mayumi Idei, Kaori Saito, Yuki Horiuchi, Kotoko Yamatani, Suzuka Ishikawa, Takamasa Yamamoto, Gene Igawa, Masanobu Hinata, Katsuhiko Kadota, Taro Kurosawa, Sho Takahashi, Takumi Saito.

**Formal analysis:** Mitsuru Wakita, Mayumi Idei, Kaori Saito, Yuki Horiuchi, Kotoko Yamatani, Suzuka Ishikawa, Takamasa Yamamoto, Gene Igawa, Masanobu Hinata, Katsuhiko Kadota, Taro Kurosawa, Sho Takahashi, Takumi Saito, Chihiro Akazawa, Tomohiko Ai.

**Investigation:** Suzuka Ishikawa, Katsuhiko Kadota, Taro Kurosawa, Sho Takahashi, Takumi Saito, Shigeki Misawa, Chihiro Akazawa, Toshio Naito.

**Project administration:** Shigeki Misawa, Toshio Naito, Yoko Tabe.

**Supervision:** Shigeki Misawa, Chihiro Akazawa, Toshio Naito, Takashi Miida, Kazuhisa Takahashi, Yoko Tabe.

**Validation:** Shigeki Misawa, Chihiro Akazawa, Toshio Naito, Takashi Miida, Kazuhisa Takahashi, Tomohiko Ai, Yoko Tabe.

**Writing – original draft:** Mayumi Idei, Yuki Horiuchi, Kotoko Yamatani, Takamasa Yamamoto, Gene Igawa, Masanobu Hinata, Tomohiko Ai.

**Writing – review & editing:** Tomohiko Ai, Yoko Tabe.

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
