## [Decision Letter · Decision Letter 0]

15 Jan 2021

PONE-D-20-41114

Comparison of the clinical performance and usefulness of five SARS-CoV-2 antibody tests

PLOS ONE

Dear Dr. Ai,

Thank you for submitting your manuscript to PLOS ONE. After careful consideration, we feel that it has merit but does not fully meet PLOS ONE’s publication criteria as it currently stands. Therefore, we invite you to submit a revised version of the manuscript that addresses the points raised during the review process.

As stated many report have explored these test, a short update of the most recent litterature in the discussion conclusion deserved to be added in addition to the minor comment from the reviewer.

We look forward to receiving your revised manuscript.

Kind regards,

Pierre Roques, Ph.D.

Academic Editor

PLOS ONE

Journal Requirements:

2. In the ethics statement in the manuscript and in the online submission form, please provide additional information about the patient records/samples used in your retrospective study, including:

a) whether all data were fully anonymized before you accessed them;

b) the date range (month and year) during which patients' medical records/samples were accessed.

Reviewers' comments:

Reviewer's Responses to Questions

**Comments to the Author**

1. Is the manuscript technically sound, and do the data support the conclusions?

Reviewer #1: Yes

2. Has the statistical analysis been performed appropriately and rigorously? 

Reviewer #1: Yes

3. Have the authors made all data underlying the findings in their manuscript fully available?

Reviewer #1: Yes

4. Is the manuscript presented in an intelligible fashion and written in standard English?

Reviewer #1: Yes

5. Review Comments to the Author

Reviewer #1: This is a study that evaluates the Roche anti-SARS-CoV-2 assay and 4 immunocromatographic assays during a short post-infection time. The study is not novel nor unique, but in the pandemic era every study dealing with findings on diagnostics of COVID-19 is valuable. My recommendation is minor revision:

- Page 5/6 The characteristics of the Roche assay are repeated, please merge this to avoid repetitions

6. PLOS authors have the option to publish the peer review history of their article (what does this mean?). If published, this will include your full peer review and any attached files.

Reviewer #1: No

---

## [Author Response · Author response to Decision Letter 0]

19 Jan 2021

Comments by the editor: Thank you for submitting your manuscript to PLOS ONE. After careful consideration, we feel that it has merit but does not fully meet PLOS ONE’s publication criteria as it currently stands. Therefore, we invite you to submit a revised version of the manuscript that addresses the points raised during the review process.

As stated many report have explored these test, a short update of the most recent litterature in the discussion conclusion deserved to be added in addition to the minor comment from the reviewer.

Answer: Thank you for the editor’s constructive comments to improve our manuscript. We cited the papers published in December 2020 and modified the texts accordingly (Page 3, ll. 55-60; Page 12-14, ll. 236-266, 267).

Comments by Reviewer #1: 

This is a study that evaluates the Roche anti-SARS-CoV-2 assay and 4 immunocromatographic assays during a short post-infection time. The study is not novel nor unique, but in the pandemic era every study dealing with findings on diagnostics of COVID-19 is valuable. My recommendation is minor revision:

- Page 5/6 The characteristics of the Roche assay are repeated, please merge this to avoid repetitions

Answer: Thank you for reviewing our manuscript and providing positive response. We modified the texts accordingly.

---

## [Editor Report · Decision Letter 1]

21 Jan 2021

Comparison of the clinical performance and usefulness of five SARS-CoV-2 antibody tests

PONE-D-20-41114R1

Dear Dr. Ai,

We’re pleased to inform you that your manuscript has been judged scientifically suitable for publication and will be formally accepted for publication once it meets all outstanding technical requirements.

Kind regards,

Pierre Roques, Ph.D.

Academic Editor

PLOS ONE
---

## [Editor Report · Acceptance letter]

29 Jan 2021

PONE-D-20-41114R1 

Comparison of the clinical performance and usefulness of five SARS-CoV-2 antibody tests 

Dear Dr. Ai:

I'm pleased to inform you that your manuscript has been deemed suitable for publication in PLOS ONE. Congratulations! Your manuscript is now with our production department. 

Kind regards, 

on behalf of

Dr. Pierre Roques 

Academic Editor

PLOS ONE